# Functional Characteristics of the Nav1.1 p.Arg1596Cys Mutation Associated with Varying Severity of Epilepsy Phenotypes

**DOI:** 10.3390/ijms25031745

**Published:** 2024-02-01

**Authors:** Grzegorz Witkowski, Bartlomiej Szulczyk, Ewa Nurowska, Marta Jurek, Michal Pasierski, Agata Lipiec, Agnieszka Charzewska, Mateusz Dawidziuk, Michal Milewski, Szymon Owsiak, Rafal Rola, Halina Sienkiewicz Jarosz, Dorota Hoffman-Zacharska

**Affiliations:** 1First Department of Neurology, Institute of Psychiatry and Neurology, Sobieskiego 9, 02-957 Warsaw, Poland; sowsiak@ipin.edu.pl (S.O.); hjarosz@ipin.edu.pl (H.S.J.); 2Military Institute of Aviation Medicine, Krasinskiego 54/56, 01-755 Warsaw, Poland; rafal.rola@wiml.waw.pl; 3Chair and Department of Pharmacotherapy and Pharmaceutical Care, Faculty of Pharmacy, The Medical University of Warsaw, Banacha 1b, 02-097 Warsaw, Poland; bartlomiej.szulczyk@wum.edu.pl (B.S.); ewa.nurowska@wum.edu.pl (E.N.); michalpasierski@gmail.com (M.P.); 4Department of Medical Genetics, Institute of Mother and Child, Kasprzaka 17a, 01-211 Warsaw, Poland; marta.jurek@imid.med.pl (M.J.); agnieszka.charzewska@imid.med.pl (A.C.); michal.milewski@imid.med.pl (M.M.); dorota.hoffman@imid.med.pl (D.H.-Z.); 5Clinic of Pediatric Neurology, Institute of Mother and Child, Kasprzaka 17a, 01-211 Warsaw, Poland; agata.lipiec@imid.med.pl; 6Institute of Genetics and Biotechnology, Faculty of Biology, University of Warsaw, Miecznikowa 1, 02-096 Warsaw, Poland

**Keywords:** voltage gated sodium channels, *SCN1A* mutation, epilepsy, patch clamp, nerve excitability study

## Abstract

Mutations of the *SCN1A* gene, which encodes the voltage-dependent Na^+^ channel’s α subunit, are associated with diverse epileptic syndromes ranging in severity, even intra-family, from febrile seizures to epileptic encephalopathy. The underlying cause of this variability is unknown, suggesting the involvement of additional factors. The aim of our study was to describe the properties of mutated channels and investigate genetic causes for clinical syndromes’ variability in the family of five *SCN1A* gene p.Arg1596Cys mutation carriers. The analysis of additional genetic factors influencing *SCN1A*-associated phenotypes was conducted through exome sequencing (WES). To assess the impact of mutations, we used patch clamp analysis of mutated channels expressed in HEK cells and in vivo neural excitability studies (NESs). In cells expressing the mutant channel, sodium currents were reduced. NESs indicated increased excitability of peripheral motor neurons in mutation carriers. WES showed the absence of non-SCA1 pathogenic variants that could be causative of disease in the family. Variants of uncertain significance in three genes, as potential modifiers of the most severe phenotype, were identified. The p.Arg1596Cys substitution inhibits channel function, affecting steady-state inactivation kinetics. Its clinical manifestations involve not only epileptic symptoms but also increased excitability of peripheral motor fibers. The role of Nav1.1 in excitatory neurons cannot be ruled out as a significant factor of the clinical phenotype.

## 1. Introduction

Voltage-dependent sodium channels (NaVs) are key regulators of cellular excitability responsible for initiation and propagation of action potential in various excitable cells, including nerve, muscle and neuroendocrine cell types. They are organized as complexes of α and β subunits, with α subunits (NaV1.x) forming sodium-permeable pores [1]. Mutations in the genes encoding these subunits—*SCNxA*—underlie several human hyperexcitability diseases, categorized as channelopathies.

The *SCN1A* gene pathogenic variants are identified in the spectrum of disorders from the most severe non-Dravet DEEs (DEE6B, MIM 619317) Dravet syndrome (DRVT/DS, MIM 607208) to milder syndromes, such as generalized epilepsy with febrile seizures plus type 2 (GEFS+ 2/GEFSP2, MIM 604403). Loss-of-function (LOF) variants in Nav1.1 cause DRVT and GEFSP2, while gain-of-function (GOF) changes have been associated with non-Dravet DEE phenotypes [2,3,4].

The p.Arg1596Cys missense variant in the *SCN1A* gene was initially reported in 2007 in a 6-year-old boy with an epilepsy phenotype within the GEFSP2 spectrum [4]. Later, Depienne and colleagues described one patient with a classical presentation of DRVT. Furthermore, [5,6] reported on a patient with atypical DRVT syndrome with ataxia and moderate intellectual disability, and Møller described a patient with GEFSP2 [7]. Finally, Hoffmann-Zacharska presented a family of three generations, in which five of the members were mutation carriers. One of them was asymptomatic, and the other four showed symptoms of the disease. Their phenotypes varied significantly in severity, from milder cases with febrile seizures (FSs) or FS+ to more severe epilepsy cases (such as epilepsy with generalized tonic–clonic seizures, focal epilepsy, atypical Dravet syndrome and Panayiotopoulos syndrome) [8].

In the present work, we invited a family of p.Arg1596Cys mutation carriers (five subjects) with different epilepsy phenotypes to participate in this study with the intention to search for potential genetic causes of symptom variability and to investigate the pathophysiological meaning of the mutation. We utilized a triple-stage approach:

(1) We performed a comparative analysis of exome sequences of mutation carriers. The WES study was performed on all affected members of the same family. We did not limit WES studies to ion channel genes but extended them to genes of other proteins involved in various functions and processes occurring in neurons that are important for their organization, structure and excitability.

(2) The patch clamp technique was employed to examine biophysical changes in the function of the ion channel caused by the mutation. This technique was also used to assess the impact of the Hm1a activator on the magnitude of sodium currents in the mutated channel, as Hm1a may, due to its selectivity, have potential therapeutic applications in the future.

(3) We also utilized a nerve excitability study (NES)—the only method that allows for an indirect assessment of ion channel kinetics in vivo [9,10]—to investigate the impact of the mutation on peripheral nerve excitability and to determine whether such changes could differentiate patients from a healthy control group. An NES is limited to the examination of peripheral nerves, which are different from central nervous system subtypes of voltage-gated Na^+^ channels (mainly Nav 1.6 [11]). Despite this limitation, the current literature provides evidence that even in channelopathies with central nervous system symptoms, like seizures in the case of voltage-dependent Na^+^ channel β-subunit mutation [9] and episodes of ataxia caused by Kv1.1 fast potassium channel dysfunction [12,13,14], complex changes in excitability can be studied using an NES and that mutation carriers can be distinguished from healthy individuals. NESs provide a valuable insight into the complex adaptative changes occurring in the nervous system as a result of the mutation.

## 2. Results

### 2.1. Patch Clamp Recordings

Sodium current activation curves were constructed using depolarization steps from −50 mV to +30 mV in 10 mV increments (Figure 1Aa,Ba). A holding potential of −90 mV was applied throughout the study.

Example recordings of the sodium current IV relationships from control and mutated channels are shown in Figure 1Ab,Bb (only sweeps evoked by −50 mV, −30 mV and −10 mV depolarization steps are shown for clarity). Full IV curves for control and mutated sodium channels are shown in Figure 1Ca. Maximal sodium currents (evoked by the 0 mV depolarization step) were much smaller when the recordings were performed for mutated sodium channels (0.17 ± 0.02 nA) as compared to the control (0.7 ± 0.15 Figure 1Ab,Bb,Ca, n = 5, *p* < 0.05).

The IV relationships were converted to conductance–voltage relationships, normalized and fitted with a Boltzmann equation to assess sodium current activation properties (Figure 1Cb).

The half-activation potentials were not different in mutated channels (−12.0 ± 2.0 mV) compared to the control (−11.6 ± 2.7 mV, n = 6, *p* > 0.05, Figure 1Cb). Additionally, the K constants of activation were not changed after mutagenesis (7.9 ± 0.2) compared to the control (7.3 ± 0.8, n = 6 *p* > 0.05, Figure 1Cb).

The voltage-gated sodium currents that were recorded in this study were fully inhibited by TTX (0.5 µM) in the extracellular solution.

Steady-state inactivation was assessed using a depolarization step that evoked the maximal current. This depolarization step was preceded by prepulses ranging from −100 mV to +20 mV that lasted 2000 ms (Figure 2Aa).

The inactivation curves were normalized and fitted with a Boltzmann equation [15]. The half-inactivation potentials were more hyperpolarized when recordings were performed from mutated sodium channels (−54.3 ± 7.4 mV) compared to the control (−39.3 ± 2.4 mV, n = 6, Figure 2Ab, *t*-test, *p* < 0.01). The K constant of the steady-state inactivation in the control (11.0 ± 1.0) was not significantly different than that after mutation (16.6 ± 4.1, n = 6, *t*-test, *p* > 0.05, Figure 2Aa).

Recovery from inactivation was assessed using two pulses evoking maximal currents with geometrically increasing interpulse intervals (from 0.5 ms to 13.5 ms, Figure 2Ba). The recovery from inactivation curves was normalized and fitted with an exponential function (Figure 2Bb).

The reactivation time constant was the same in the control (11.5 ± 0.7 ms) and in the presence of a mutation (10.5 ± 1.5 ms, n = 5, Figure 2Bb, *t*-test, *p* > 0.05).

Moreover, the influence of the selective sodium channel activator Hm1a 500 nM on the maximal control and mutated sodium channels was assessed. The maximal amplitude of control sodium channels was not influenced by Hm1a (0.34 ± 0.13 nA and 0.33 ± 0.11 nA before and after application of activator, respectively, Figure 3Aa, n = 7, *t*-test, *p* > 0.05). Similarly, the maximal amplitude of mutated sodium channels was not influenced by Hm1a (0.14 ± 0.04 nA and 0.15 ± 0.04 nA before and after application of activator, respectively, Figure 3Ba, n = 6, *t*-test, *p* > 0.05). Hm1a, however, potently prolonged the time-dependent inactivation of control and mutated sodium channels. In control channels, the tau of time-dependent inactivation was 0.64 ± 0.08 and 1.36 ± 0.22 before and after Hm1A application, respectively (Figure 3Aa,Ab, n = 8, *t*-test, *p* < 0.05). Similarly, in mutated channels, the tau of time-dependent inactivation was 0.69 ± 0.21 and 2.7 ± 0.69 before and after Hm1a application, respectively (Figure 3Bb, n = 5, *t*-test, *p* < 0.05).

### 2.2. Nerve Excitability Study

Preliminary nerve conduction study (NCS).

In all patients, an NCS performed before the nerve excitability study ruled out the presence of median nerve injury due to potential carpal tunnel syndrome (see Table 1).

Strength–duration time constants.

*SCN1A* mutation carriers were characterized by significantly lower rheobase when compared to the control group (3.87 ± 0.44 mA, n = 5 vs. 7.97 ± 1.76 mA, n = 10, respectively, *t*-test, *p* = 0.038), with almost equal strength–duration time constants (0.37 ± 0.02 ms vs. 0.4 ± 0.03 ms, respectively, *t*-test, *p* = 0.53). These values are reflected in Figure 4A, where the line fitted to the charge–duration plots represents the medium value for mutation carriers characterized by a much shallower trajectory compared to the control group.

Recovery cycle.

The relative refractory period was found to be significantly shorter in subjects possessing *SCN1A* mutations when compared to the control group (2.88 ± 0.09 ms n = 5 vs. 3.42 ± 0.18 ms n = 10, *t*-test, *p* = 0.045, Figure 4B). However, refractoriness at 2 and 2.5 ms did not reach statistical significance, despite a trend towards lower values that was present in *SCN1A* mutation carriers (see Table 1).

Threshold electrotonus.

There were no significant changes during the depolarization phase of the threshold electrotonus between the data recorded for patients and for the control group (Figure 4C; for excitability changes (Td) in all time intervals, see Table 1). However, in the hyperpolarizing part of the threshold electrotonus, there was a significantly greater decrease in membrane excitability (so-called fanning out, see [16]) found in subjects with a mutation compared to the control group and seen at both −20% and −40%, progressing during the time course of the hyperpolarizing pulse. For the time interval of 90–100 ms of −40%, the hyperpolarizing pulse excitability change was −130.37% ± 7.3 n = 5 for mutation carriers and −83.9% ± 11.65 for the control group; n = 10, *p* = 0.041, *t*-test (see Table 1 for the remaining excitability parameters during hyperpolarization).

Current–voltage relationship.

The hyperpolarizing I/V slope recorded for subjects with *SCN1A* mutation was characterized by a trend towards less inward rectification (the part of the curve below the x axis shifted left) compared to recordings from the control group. (Figure 4D, the slope 0.324 ± 0.03 ν = 5 vs. 0.299 ± 0.021, *p* = 0.55, *t*-test). Although this result did not reach statistical significance, it is in line with the results that suggest a more pronounced hyperpolarizing threshold electrotonus in patients with *SCN1A* mutations.

### 2.3. WES Analysis

Three variants present in Proband and absent in other family members, which could act as phenotype modifiers, were found (Table 2); however, their clinical significance remains uncertain.

## 3. Discussion

It is known that approximately 70–85 percent of patients with Dravet syndrome have a mutation in the *SCN1A* gene, although the clinical manifestations associated with a given mutation may result in a diverse clinical phenotype. Our research, conducted using three different techniques, focused on determining the characteristics of sodium currents of the Nav1.1 channel with a p.Arg1596Cys mutation, as well as on attempts to explain the causes of phenotypic diversity in people with the p.Arg1596Cys mutation. The NES technique enabled the study of motor neurons in vivo, extending the characterization of clinical symptoms.

### 3.1. Patch Clamp Findings

Our patch clamp recordings focused on the biophysical properties of the p.Arg1596Cys channel. Our results indicate that the mutated channel generates a sodium current with an amplitude that is about 75% lower than that of the WT channel. The inactivation curve shifted towards hyperpolarization potentials with no changes in the kinetics of recovery from inactivation. The activation properties of the p.Arg1596Cys channel did not differ from the properties of the WT channel. The shift in the inactivation towards hyperpolarization with no shift in the activation indicates that the “window current”, which is determined by the crossover of the activation and inactivation curves, was reduced. We conclude that the overall effect of the p.Arg1596Cys mutation on neuron excitability was inhibitory, while the ability to conduct sodium current was preserved. Therefore, our results differ from previous research [17]. Due to the severity of clinical symptoms in DS, one would expect that the variants responsible for the disease would show the biophysical characteristics of severe LOF defects. On the other hand, the presence of asymptomatic and mild disease carriers may indicate that the mutant channel partially retains its function. Our observation that the current persists despite the mutation is consistent with the fact that p.Arg1596Cys is a missense mutation. It has been suggested that missense mutations typically modify the channel function rather than abolish its function [18]; however, in the case of *SCN1A* LOF, mutations missenses are nearly equally as common as truncating mutations.

From a therapeutic point of view, the susceptibility of the mutant channel to the action of Hm1a—a spider venom component—gives hope for finding a therapy based on the mechanism of prolonging the time-dependent inactivation. This result is confirmed by recent findings demonstrating that HM1a mediated selective Nav1.1 activation in mice harboring an *Scn1a* LOF mutation, leading to potent suppression of seizure activity [19].

It should be noted that there are also other new therapeutic options that could be used to treat epilepsy caused by Nav1.1 sodium channel mutations. One of them is pharmacological modulation of the K+Cl-cotransporter [20]. Other methods include responsive neurostimulation, vagus nerve stimulation and deep brain stimulation [16]. Dietary interventions can also be considered [21].

### 3.2. NES Examination Findings

The nerve excitability studies performed in patients possessing *SCN1A* mutations revealed two significant findings: almost 2× lower rheobase levels and a shorter relative refractory period when compared to healthy subjects. These findings suggest an increased excitability of peripheral motor fibers caused by changes in both quickly and slowly inactivating subgroups of voltage-dependent Na^+^ channels. Rheobase is influenced mainly by a subpopulation of slowly inactivating or persistent sodium currents [22] active at near-threshold membrane potentials and responsible for increased membrane conductance for Na^+^ ions, leading to the depolarization of the membrane. Thus, low rheobase results in increased conductance for sodium, which may be attributed to either a higher subpopulation of Na^+^ channels working in the “persistent” mode or an increased Na^+^ gradient throughout the membrane. The RRP is shaped by fast or time-dependent inactivation of Na^+^ channels, and its shortening produces a faster generation of subsequent action potentials. The same pattern of hyperexcitability of motor neurons was observed in all patients with *SCN1A* mutation.

Although the Nav1.1 channel was formerly known as the brain type I channel, its presence is not isolated to the brain or to the CNS. Nav1.1 is expressed within nodes of Ranvier and axon initial segments (AISs) throughout the mouse spinal cord [23].

In the PNS, the presence of Nav1.1 is confirmed in the dorsal root ganglions (DRGs) and retinal ganglion cells. The presence of Nav1.1 in DRGs was shown by [24,25]. If we assume that Nav1.1 is present in the DRGs and nodes of Ranvier in the gray matter, we can expect that the transmission of sensory signals reaching motor neurons in subjects with the mutation will be weaker than those in healthy people. Persistently weakened transmission likely activates compensatory mechanisms that increase the excitability of motor neurons in order to maintain their activity at an appropriate level. The increased excitability of motor neurons in patients with *SCN1A* mutations in NES tests may therefore be explained by such compensatory mechanisms. Recent findings in this field suggest novel mechanisms of adaptative changes, e.g., increased expression of Activity Regulated Cytoskeleton Associated Protein (Arc) [26]. We also speculate that increased Nav1.6 AIG expression in motoneurons in *SCN1A* mutation carriers may compensate for Nav1.1 channel malfunction. Also, from a practical point of view, significant differences in rheobase and the RRP may suggest that NESs can be used in differential diagnostics of *SCN1A* channelopathies before genetic testing.

A serious limitation of the clinical arm is the fact that all patients with investigated mutations are currently being successfully treated with the antiepileptic drug valproic acid. Therefore, we cannot exclude the impact that adaptive changes may have had on the results for excitability. However, valproic acid suppresses seizure activity in different ways—mostly via the GABAergic mechanism—and its influence on the kinetics of voltage-gated sodium channels is relatively small [27] and limited to the TTX-resistant fraction of the current [28]. It was of course impossible to perform the study on untreated subjects.

### 3.3. Why Does Reduced Nav1.1 Sodium Current Cause Epilepsy?

Based solely on the decreased current amplitude and steady-state properties of Na^+^ channel inactivation in a whole-cell patch clamp model, it can be assumed that neurons expressing the dysfunctional channel show anomalous inactivation and decreased excitability. However, alterations in the biophysical properties observed in the Arg1596Cys channel did not seem to correspond directly to the symptoms of epilepsy nor to the hyperexcitability observed in peripheral nerves in the NES tests. This discrepancy can be explained by the fact that the sodium channel protein Nav1.1, encoded by the *SCN1A* gene, is found mainly in CNS inhibitory GABAergic interneurons. Reducing their activity results in an imbalance of excitatory over inhibitory electrical signaling [29,30]. Therefore, the functional compromise of this subgroup of cells may lead to the severe disinhibition of cortical glutaminergic neurons, which may help to explain the development of epileptic disorders in carriers of this mutation.

The research of Ogiwara et al. [31] indicates that in addition to the dominant expression in GABAergic inhibitory neurons, Nav1.1 channels are expressed in subpopulations of excitatory neurons, including entorhinal-hippocampal projection neurons, a subpopulation of neocortical layer V excitatory neurons and thalamo-cortical projection neurons. They showed that Nav1.1 haploinsufficiency in excitatory neurons has an ameliorating effect on the pathology of Dravet syndrome in a mouse model. Therefore, variability in the epilepsy phenotype may be related to the expression of the Nav1.1 channel in the subpopulations of excitatory neurons. The presence of mutated channels in excitatory neurons could compensate for the imbalance in the excitability of CNS caused by insufficient activity of sodium channels in GABAergic neurons.

If such a mechanism were appropriate, phenotypic diversity would not depend on variability in the genotype of other ion channels but rather on the mechanisms responsible for the quantitative distribution of Nav1.1 channels in excitatory neurons. The proteins that coordinate the distribution of Nav1.1 channels in different cell types are not yet known. Therefore, we conducted WES studies to understand the general spectrum of gene variability in the hope of clarifying whether there is a factor common to the studied group of mutation carriers that could explain the mechanism causing phenotypic variability.

### 3.4. WES Findings

The diversity of phenotypes associated with Arg1596Cys missense variants suggests that there may be other dysfunctional proteins encoded by different genes that modulate sodium channel activity. In 2013, Ohmori and colleagues showed that patients carrying mutations in *CACNA1A*, a gene encoding the Cav 2.1 calcium channel, in addition to mutations in *SCN1A*, had more absent seizures, earlier onset of seizures and more prolonged seizures [32]. In their analysis of 109 DRVT patients, Singh et al. [33] found 9 subjects with mutations in the *SCN9A* gene coding for the Nav 1.7 channel protein, 6 of which also had mutations in the *SCNA1A* gene [33]. It has been shown that individuals with genetic epilepsy usually carry more than one variant in genes coding for ion channels. This was the reason why we conducted a comparative analysis of the WES data of a patient with DRVT (Subject No. 2, see Table 3) as well as considering the data of other family members with a milder phenotype. We did not find any unequivocally pathogenic variants that could be the cause of the disease phenotype (severe epilepsy and autistic features) in Proband. We found three variants that were present in Proband and absent in other family members that could act as phenotype modifiers, but their clinical significance is uncertain. We found a missense variant, c.4772G>A, in the *CACNA1H* gene. The *CACNA1H* gene mutations encoding the α1H subunit of the Cav3.2 T-type calcium channel is associated with susceptibility to generalized epilepsy and focal or multifocal epilepsy of varying severity, in addition to developmental delay and autism [34]. We also found a missense variant in the *KIF4A* gene. Kinesin superfamily (*KIF*) genes encode motor proteins that have fundamental roles in brain functioning, development, survival and plasticity [35]. The kinesins lead microtubule transport within axons, dendrites and synapses and affect synaptic excitability. It has been shown that mutations in *KIF4A* are related to seizures, autism and intellectual disability [36]. We also identified a pathogenic frameshift variant in one allele of the *GJB2* gene. *GJB2* encodes connexin 26, a gap junction protein with a crucial role in neuronal migration. Mutations in *GJB2* are responsible for recessive and dominant forms of deafness, but it was shown that the phenotypic spectrum of *GJB2* mutations could be expanded to include epileptic manifestations [37]. The phenotypes evoked by these three genes could partially explain the disease worsening in Proband, but it is difficult to determine their actual impact without functional studies.

## 4. Materials and Methods

### 4.1. Patch Clamp

#### 4.1.1. Cell Culture and Transfection

Human kidney cells, HEK293T (Sigma-Aldrich, Burlington, MA, USA), were cultured as described previously [38] in DMEM supplemented with heat-inactivated fetal calf serum (10%), l-glutamine (2 mM), penicillin (100 units/mL) and streptomycin (100 μg/mL). Cell culture reagents were from Gibco (Waltham, MA, USA). One day before transfection, cells were seeded into silicon culture inserts (Culture-Inserts 2 Well form ibidi) on glass coverslips. Plasmids encoding cDNA *SCN1A* WT (NM_006920; OriGene RG220167) and its derivative, p.Arg1596Cys (AB093548.1; acc. NM_006920 c.5781C>T, p.Arg1585Cys), were transiently expressed in HEK293T cells using the FuGENE6 Transfection Reagent (Promega, Madison, WI, USA). Experiments were performed 48 h after transfection. A total of 2 μg of plasmid DNA for 8 μL FuGENE6 was used for 0.88 cm^2^ of growth area.

#### 4.1.2. Patch Clamp Recordings

Control and mutated Nav1.1 voltage-gated sodium channels (mutation p.Arg1596Cys) expressed in HEK cells were recorded using the patch clamp technique.

Cells were visualized using a Nikon inverted microscope (NIKON Diaphot 300, Tokyo, Japan). Nav1.1-tGFP-expressing HEK-293T cells were detected by GFP fluorescence. Currents were evoked using rectangular voltage steps lasting 20 msec from the holding potential of –90 mV.

The intracellular solution contained (in mM) CsF (95), CsCl (13), NaCl (7), EGTA (1), HEPES-Cl (10) and MgCl_2_ (2) at pH 7.3 and osmolarity 280 mOsm.

The extracellular recording solution contained the following components (in mM): NaCl (149), KCl (3), TEA-Cl (10), CaCl_2_ (2), MgCl_2_ (2), glucose (5), HEPES (10) and CdCl_2_ (0.2) at pH 7.4.

The electrophysiological techniques used were the same as those employed in our previous study [32]. Patch pipettes had resistances between 4 and 5 MΩ. The pipette was moved towards a cell using a micromanipulator (Sutter Instruments, Novato, CA, USA). After gigaseal formation, the electrode capacitance was compensated for. The patch membrane was ruptured by suction and/or by an electrical stimulus. After that, the membrane capacitance was compensated for. The access resistance was between 5 and 7 MΩ. A series resistance compensation of 80% was applied, and the P-4 protocol was enabled. Recordings were performed at 35–36 °C using an amplifier, Axopatch 1D, and a Digidata 1550B (Molecular Devices, San Jose, CA, USA). Voltage-gated calcium and potassium currents were not recorded because they were not expressed in HEK293T cells. Moreover, calcium and potassium channels were blocked by cadmium ions and TEA-Cl, respectively, in the extracellular solution. The patch clamp methodology is shown on Figure 5.

All patch clamp results are shown as mean ± SE (paired or unpaired Student’s *t*-test was used, GraphPad InStat software v3.06, GraphPad, La Jolla, CA, USA).

### 4.2. Nerve Excitability Study

#### 4.2.1. Clinical Characteristics of the Participants

The NESs were performed in 5 family members with the presence of the p.Arg1596Cys mutation. Patients were identified based on records from a pediatric neurological outpatient clinic, and the results of genetic testing performed at the Department of Genetics, Institute of Mother and Child, were used. All patients were free of epileptic seizures for more than a year but remained on antiepileptic treatment. All patients received sodium valproate in doses between 600 and 1800 mg per day to maintain a therapeutic concentration in the blood. There were no signs or symptoms of other metabolic or systemic disorders in their medical history. Patients were diagnosed in the Department of Clinical Genetics, Institute of Mother and Child, and remained under the care of the neurology outpatient clinic. The detailed clinical data are presented in Table 3.

We also performed an NES in 10 healthy volunteers—the control group recruited by the patients’ caregivers and hospital workers.

#### 4.2.2. The Methodology of Nerve Excitability Study (NES)

The NES was performed with the use of the standard setup for a basic nerve conductance study [39]. The stimulating electrode was located above the median nerve at the wrist, and the response electrode recording the compound action potential (CAP) was placed above the abductor pollicis brevis (ABP) muscle. The stimulating electrode was connected to a stimulator DS5 (Digitimer, Letchworth Garden City, United Kingdom) controlled by Qtrac software ver 17/5/2017 (Digitimer, Letchworth Garden City, United Kingdom) installed on a PC computer. The recording electrode was connected to a standard EMG device (Nicolet EDX Viking, Nicolet Biomedical Inc, Madison, WI, USA) linked to a computer through a digital-to-analog converter (NI DAQ 6221, National Instruments, Austin, TX, USA) (Figure 5). Recordings were performed with the use of the TROND protocol utilizing the technique of threshold tracking, where a set of long subthreshold depolarizing and hyperpolarizing pulses (prepulses) was applied to the nerve, and the system tracks how the nerve’s excitability changes in response to prolonged depolarizations and hyperpolarizations. The NES setup is presented in Figure 6. With the use of the TROND protocol, a series of excitability parameters were measured [10]:The strength–duration relationship between the duration and strength of pulse necessary to evoke a compound muscle-evoked potential (MEP) with a preset amplitude (40% of maximum) was calculated automatically with the use of the Weiss empirical law, Q=Ixt = Irh (t + SDTC), where Q is the stimulus charge, I is the stimulus current of duration t, SDTC is the strength–duration time constant, and I_rh_ is the rheobasic current. Measurements revealed the nerve fiber rheobase (the lowest current with an infinite duration that induces a response, calculated as a slope of the straight line fitted to the points on the charge–stimulus duration plot), and chronaxie (equal to the strength–duration time constant), found as the x intercept of the above-mentioned straight line [40]. Rheobase is influenced mainly by the activity and properties of persistent sodium currents generated in the nodes [22].The subprotocol for the threshold electrotonus contained four series of subthreshold depolarizing and hyperpolarizing prepulses set to +/− 20 and +/− 40% of the unconditioned threshold current, lasting for different durations (between 1 and 200 ms). The test pulse generated after each prepulse was automatically adjusted to reach the preset amplitude of the fiber response. The threshold electrotonus curve represents membranes’ behavior during prolonged de- and hyperpolarizations and is shaped by the activity of voltage-dependent sodium and voltage-dependent slow potassium and inward rectifying channels [41].The recovery cycle measured membrane excitability changes after the generation of the compound action potential. The protocol tracks 3 phases of returning to normal excitability: the relative refractory period (RRP), the superexcitable period and the late subexcitability period. The RRP reflects the recovery of inactivated voltage-dependent sodium channels in the nodes [42], where the superexcitability is influenced by prolonged internodal depolarization and late subexcitability by the slow kinetics of voltage-dependent potassium channels from the afterhyperpolarization [43].The current–threshold relationship reveals the dependence between the different levels of fiber resting potential shaped by subthreshold depolarizing and hyperpolarizing pulses with a fixed length and the current necessary to evoke the CAP with a fixed amplitude. The current–threshold curve is sensitive to processes modifying membrane polarization; during hyperpolarizing pulses, the activity of inward rectifying potassium channels limits the decrease in membrane excitability [10].

In all cases, the NES was preceded by a short nerve conduction study to exclude potential carpal tunnel syndrome; in the local lab, the compound motor action potential (CMAP) distal latency was considered normal when <4.2 ms (8 cm distance). Before the NES, limbs were warmed with the use of an electric pillow and surface thermometer to maintain a stable skin temperature at 32 degrees Celsius.

#### 4.2.3. Statistical Analyses

The data presented in graphs are depicted as means ± standard error (SE). The core NES data summarized in Table 2 are presented as means (range). Statistica ver. 12 (StatSoft Tulsa, OK, USA) software was used for statistical analyses. The normality of the sample distribution was measured with the Kolmogorov–Smirnov test, and Levene’s test was routinely applied to assess the homogeneity of variance between samples. Parametric variables were compared with the use of independent samples *t*-tests, and the Mann–Whitney test was used when the data did not follow a normal distribution. Differences were considered significant when *p* ≤ 0.05.

### 4.3. Exome Sequencing and Data Analysis

Exome sequencing (WES) was performed based on the Agilent Sure Select Human All Exon v6 system (Agilent Technologies, Santa Clara, CA, USA) on the HiSeq 1500 platform (Illumina, San Diego, CA, USA). The analysis of variants in clinically significant genes was performed using a data processing pipeline based on VEP algorithms developed by the Department of Medical Genetics, Institute of Mother and Child, human genome version GRCh38/hg38. The quality analysis of the obtained sequences was performed using the IGV 2.7 program (Broad Institute, Cambridge, MA, USA). WES analysis was performed in all the *SCN1A* mutation carriers in the family. The analysis was conducted for the presence of unique variants found in Proband (more severe phenotype—Subject No. 2, see Table 1) and absent in other *SCN1A* mutation carriers (milder phenotype) to find putative phenotype modifiers. We analyzed variants in genes according to panel brain channelopathy v. 1.70 and genetic epilepsy syndromes v. 2.477 (Genomics England PanelApp), genes related to channelopathies (according to different panels, 406 genes, Appendix A) and all variants classified as pathogenic in ClinVar (https://www.ncbi.nlm.nih.gov/clinvar/ accessed on 15 December 2023).

## 5. Conclusions

Overall, our findings suggest that p.Arg1596Cys substitution in the *SCN1A* gene leads to the significant inhibition of Na^+^ channel function, mainly modifying the maximal current amplitude and inactivation properties. On the clinical level, this mutation is associated with a significant increase in the excitability of peripheral motor fibers that can be detected in a nerve excitability study. On the cellular level, the results for the mutation may be partially reversed with the application of the Hm1a Na^+^ channel activator. The phenotypic differences between tested subjects may be partially explained by their genetic background and the coexistence of variants in other genes, such as *CACNA1H*, *KIF4A* and *GJB2*. However, the role of Nav1.1 channels in excitatory neurons as the main factor influencing the phenotype cannot be ruled out.

## Figures and Tables

**Figure 1 ijms-25-01745-f001:**
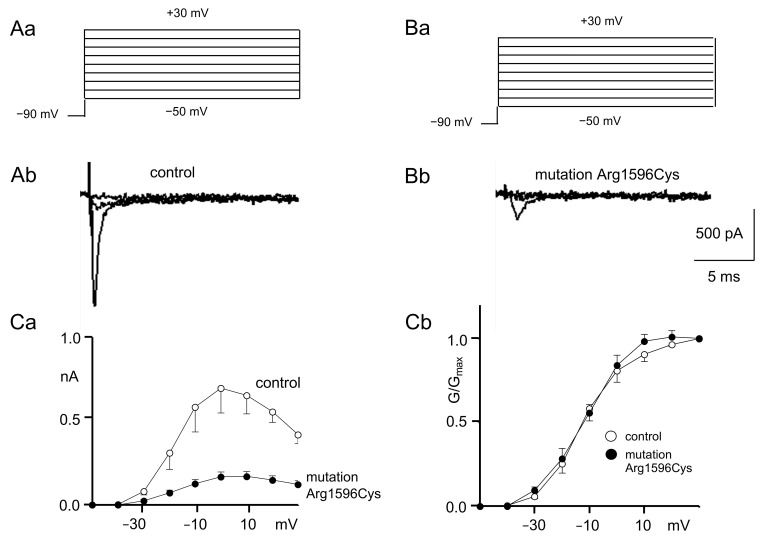
Control and mutated (Arg1596Cys) Nav1.1 sodium channel activation. (**A**) Recordings of control sodium current activation. ((**Aa**) Voltage protocol; (**Ab**) example control recordings evoked by −50 mV, −30 mV and −10 mV depolarization steps). (**B**) Recordings of mutated sodium current activation; (**Ba**) voltage protocol; (**Bb**) example recordings of mutated channels evoked by −50 mV, −30 mV and −10 mV depolarization steps). (**Ca**) IV relationships of control (white circles) and mutated (black circles) Nav1.1 sodium channels. The mutation strongly decreases current amplitudes (see Figures A and B). nA: nanoAmper. (**Cb**) Normalized conductance–voltage relationships of control (white circles) and mutated (black circles) channels.

**Figure 2 ijms-25-01745-f002:**
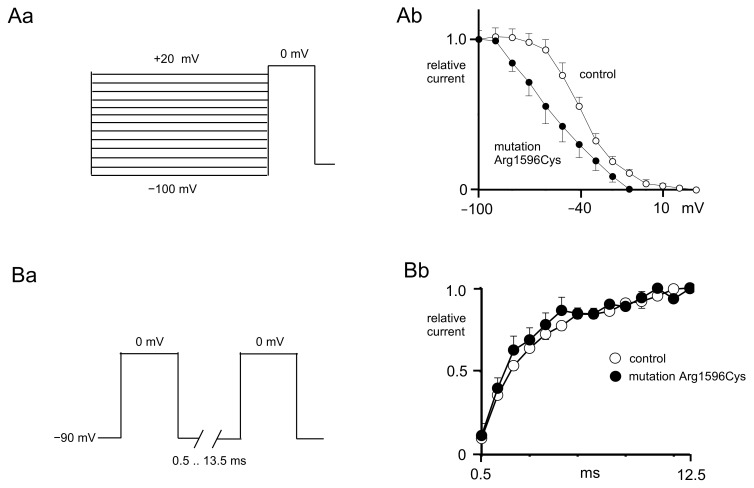
Control and mutated (p.Arg1596Cys) Nav1.1 sodium channel inactivation and reactivation curves. (**Aa**) Voltage protocol used to study steady-state inactivation of sodium currents. (**Ab**) Steady-state inactivation curve of mutated channels (black circles) shifted towards hyperpolarization as compared to control channels (white circles). (**Ba**) Voltage protocol used to study recovery from inactivation. (**Bb**) Recovery from inactivation of control (white circles) and mutated (black circles) sodium channels.

**Figure 3 ijms-25-01745-f003:**
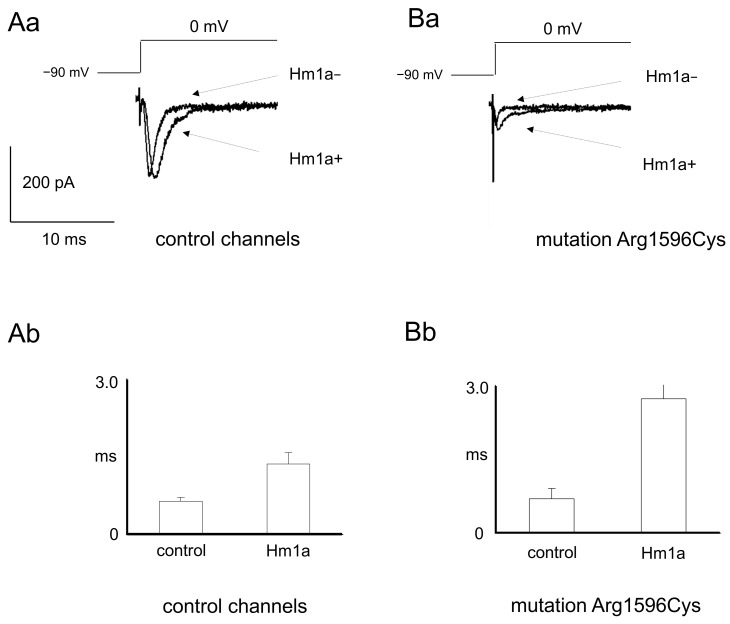
Sodium currents before and after application of sodium channel activator Hm1a in wild-type and mutated channels (p.Arg1596Cys). (**Aa**) Example recordings of control sodium currents before and after application of Hm1a. (**Ab**) Averaged time-dependent inactivation tau constants before and after application of Hm1a in control channels. ms-miliseconds. (**Ba**) Example recordings of mutated sodium currents before and after application of Hm1a. (**Bb**) Averaged time-dependent inactivation tau constants before and after application of Hm1a in mutated channels.

**Figure 4 ijms-25-01745-f004:**
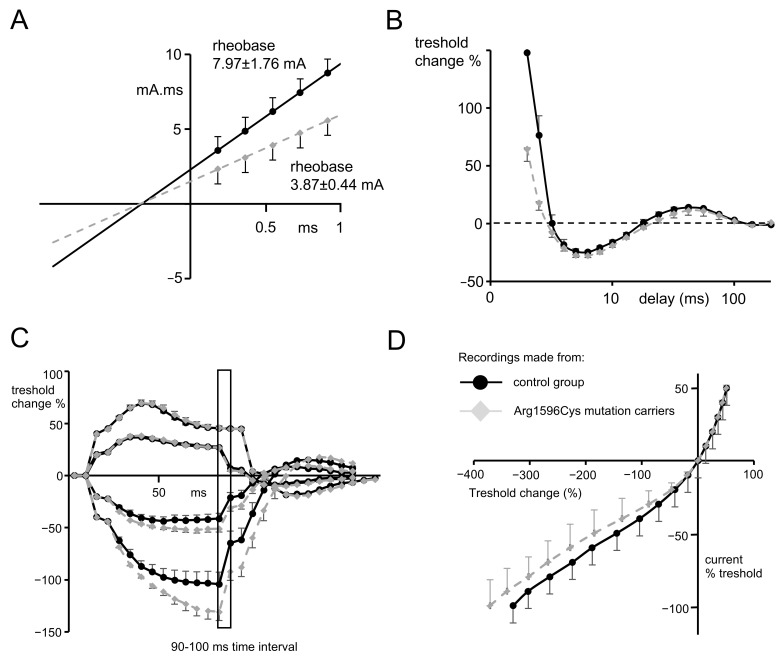
(**A**) Plot presenting threshold charge (mA.mS) versus stimulus duration, where the slope represents rheobase and the negative intercept on the *x* axis of the line that was determined by 5 consecutive stimulus widths represents chronaxie or the strength–duration time constant. For this and other graphs, the gray dots represent mean data from Arg1596Cys mutation carriers, and black dots represent the mean data from control group. (**B**) Recovery cycle plot. The first part of the waveform—above 0%—represents excitability changes during relative refractory period. ms-miliseconds. The second part—below 0%—represents those occurring during the supernormal period, and the third part, again above 0%, represents those occurring during the late subnormal period. (**C**) Threshold electrotonus induced by test pulses applied directly after depolarizing (above *x* axis) and hyperpolarizing (below *x* axis) 20% and 40% threshold conditioning currents. Hyperpolarizing waveforms recorded for mutation carriers represent widening out (or fanning out), which is characteristic of a more profound reduction in excitability during hyperpolarization. Insert of the graph represents 90–100 ms time interval, for which responses to the test stimuli have undergone statistical analysis. (**D**) Current–voltage relationship waveform representing threshold changes 200 ms after the onset of long depolarizing or hyperpolarizing currents (*x* axis) plotted against the % of polarizing current.

**Figure 5 ijms-25-01745-f005:**
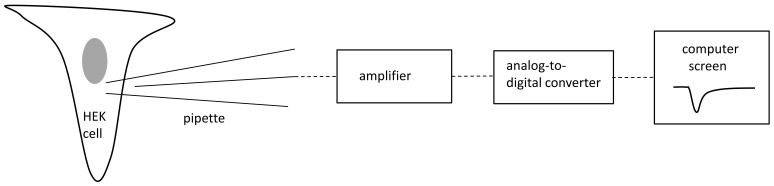
Conceptual diagram of patch clamp methodology. Grey color-nucleus.

**Figure 6 ijms-25-01745-f006:**
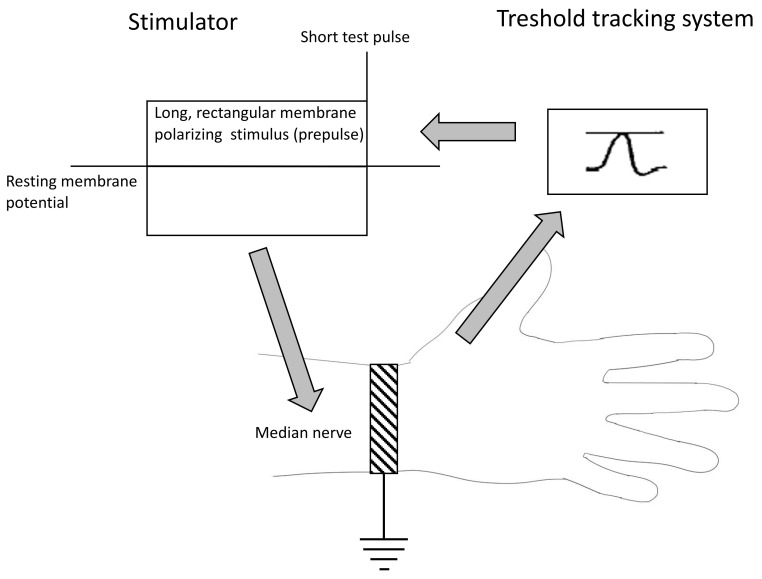
The nerve excitability study diagram.

**Table 1 ijms-25-01745-t001:** Core nerve excitability parameters in control group subjects and *SCN1A* mutation carriers. Significantly different parameters are presented in **bold**.

	Control Group		*SCN1A* Mutation Carriers		
	Mean	Range	Mean	Range	P (*t*-Test)
**Distal latency (ms)**	3.3	2.7–3.5	3.1	2.5–3.4	0.12
**Excitability parameters**					
Stimulus (mA) for 50% max response	8.85	7.11–12.94	5.64	4.17–7.1	0.09
Strength–duration\time constant (ms)	0.4	0.25–0.55	0.37	0.32–0.41	0.53
**Rheobase (mA)**	**7.97**	**5.1–13**	**3.87**	**2.75–5.13**	**0.038**
Stimulus–response\slope	6.43	5.43–8.64	5.14	3.84–5.85	0.13
Peak response\(mV)	0.53	0.31–0.67	0.63	0.27–0.78	0.5
Resting I/V slope	0.61	0.52–0.69	0.5	0.24–0.64	0.2
**RRP (ms)**	**3.41**	**2.95–3.74**	**2.88**	**2.66–3.12**	**0.045**
Superexcitability (%)	−23.8	−18.4–(−32.9)	−24.6	−18–(−34.2)	0.51
Subexcitability (%)	16.48	11.54–20.8	10.1	9.3–13.1	0.14
Refractoriness at 2.5 ms (%)	26.25	21.24–33.12	26.9	23.3–33.2	0.86
TEd (10–20 ms)	61.64	2.83	68.06	1.85	0.3
TEd (40–60 ms)	51.2	44–57	53.5	47–58	0.68
TEd (90–100 ms)	45.2	41.6–49.4	44.6	40.27–48.6	0.77
**TEh (10–20 ms)**	**−67.8**	**−66.1-(−78.2)**	**−77.2**	**−75.7-(−83.1)**	**0.03**
**TEh (20–40 ms)**	**−85.12**	**−83.6-(−96.1)**	**−96.3**	**−93.3-(−107.1)**	**0.045**
**TEh (slope 101–140 ms)**	**1.79**	**1.69–2.28**	**2.26**	**2.07–2.8**	**0.046**

Abbreviations: RRP—relative refractory period. TEd—change in nerve fiber excitability during depolarizing of the threshold electrotonus. TEh—change in nerve fiber excitability during hyperpolarizing threshold electrotonus.

**Table 2 ijms-25-01745-t002:** Gene variants related to epilepsy, unique to Proband (Subject No. 2, see Table 1), that were absent in other *SCN1A* mutation carriers in the family.

Gene	Reference Sequence	cDNA	Protein	Zygosity	Effect	gnomAD	CADD	Classification
*CACNA1H*	NM_021098.3	c.4772G>A	p.(Ser1591Asn)	het	missense	0.00006	8.171	VUS
*KIF4A*	NM_012310.5	c.2555A>T	p.(Gln852Leu)	het	missense	0.00001	21.3	VUS
*GJB2*	NM_004004.6	c.35del	p.(Gly12ValfsTer2)	het	frameshift	0.00643	-	Pathogenic

**Table 3 ijms-25-01745-t003:** Clinical characteristics of the Arg1596Cys mutation carriers.

Subject No.	1	2	3	4	5
Age at investigation	53 y	16 y	51 y	17 y	15 y
Gender	M	M	M	M	F
Prenatal history	Uneventful	Uneventful	Uneventful	Uneventful	Uneventful
Psychomotor development	Normal	Normal	Normal	Normal	Normal
Age at seizure onset	13 y	14 m	4 y	3 y	4 y
Triggering factors at onset	Fatigue	Fever	Physical effort, hyperthermia?	Fever	Fever
Age at last seizure	53 y	6 y	39 y	12 y/convulsive status epilepticus	10 v
Types of seizures	GTCS	Myoclonic, GTCS, absence, focal impaired awareness, predominantly on the right side	GTCS	GTCS	GTCS
Clusters of seizures	No	Yes	No	No	None
Convulsive status epilepticus	No	Yes	No	Yes	None
Non-convulsive status epilepticus	No	No	No	No	None
Cognitive development	Normal	Normal	Normal	Normal	Normal
ASD	No	Asperger syndrome	No	No	No
Neurological examination	Normal	Ataxia	Normal	Slight ataxia, clumsiness	Normal
EEG (interictal)	Normal	Focal, lateralized and generalized spikes, multispikes, spikes–waves predominantly on the left side	Generalized spikes–waves	Normal	Normal
Neuroimaging	MRI-normal	MRI-normal	MRI-normal	MRI-normal	MRI-normal
AED previous	CBZ	VPA; VPA + OXCB; VPA + LTG; VPA + LEV	CBZ; CBZ + VPA; OXCB + VPA; VPA	VPA	VPA
AED current	VPA 1800 mg/d	VPA 600 mg/d	VPA 1000 mg/d	VPA 700 mg/d	VPA 600 mg/d
SCN1A-related epilepsy phenotype	Epilepsy with GTCS	Dravet syndrome—atypical	Epilepsy with GTCS	GEFS(+)	Epilepsy with GTCS

Abbreviations: GTCSs—generalized tonic–clonic seizures, AEDs—antiepileptic drugs, VPA—valproic acid, OXCB—oxcarbazepine, LTG—lamotrygine, LEV—levetiracetam, ASD—autism spectrum disorder, GEFS(+)—genetic epilepsy with febrile seizures plus, MRI—magnetic resonance imaging.

## Data Availability

The data presented in this study are available on request from the corresponding author. The data are not publicly available to protect participants’ privacy.

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
