# Peer review of "Functional Characteristics of the Nav1.1 p.Arg1596Cys Mutation Associated with Varying Severity of Epilepsy Phenotypes"

_ijms, 2024, doi:10.3390/ijms25031745_

Round 1

Reviewer 1 Report

Comments and Suggestions for Authors

Manuscript Title: Functional Characteristics of the Nav1.1 p.Arg1596Cys Mutation Associated with Varying Severity of Epilepsy Phenotypes

The study by Witkowski et al. delves into the functional characteristics of the SCN1A gene mutation, shedding light on its association with diverse epilepsy phenotypes. While the paper explores an important topic, there are areas where improvements can enhance its clarity and impact.

The manuscript, while informative, can be more succinct. Streamlining the language without sacrificing clarity is advisable.

The methodology needs a more detailed explanation, particularly regarding the patch-clamp analysis and neural excitability studies. A brief schematic or concept diagram elucidating the experimental setup could aid comprehension.

Results should be presented with a focus on key findings, facilitating easier interpretation for readers.

The language used is generally clear but can benefit from a more concise and structured approach. Ensure that each sentence contributes directly to the understanding of the study. Clarify certain technical terms and abbreviations for the broader readership.

Consider incorporating a conceptual diagram illustrating the experimental design. This can enhance the paper's accessibility and aid in understanding complex methodologies.

The paper alludes to investigating genetic causes for clinical variability. Expanding on this aspect, perhaps by providing additional context or potential hypotheses, could enrich the genetic analysis section.

Some recent papers should be discussed in the discussion section: https://pubmed.ncbi.nlm.nih.gov/37704745/ ; https://pubmed.ncbi.nlm.nih.gov/37759906/ ; https://pubmed.ncbi.nlm.nih.gov/33923061/

A more explicit statement linking the functional characteristics of the mutation to the varying clinical phenotypes would reinforce the paper's main findings.

Addressing the suggested improvements can significantly enhance its clarity and accessibility, ensuring its impact on the readership. I recommend a revision before considering publication.

Comments on the Quality of English Language

Needs improvement

Author Response

Dear Reviewer,

We would like to thank you for all valuable remarks. We applied many changes to the text based on above suggestions, being absolutely sure that this will make our paper clearer and will add value to it.

Here are detailed responses to Reviewers’ remarks:

  • The manuscript, while informative, can be more succinct. Streamlining the language without sacrificing clarity is advisable.

We shortened most of sections, especially Introduction and Discussion to make the text more informative and focused on the research results.

  • The methodology needs a more detailed explanation, particularly regarding the patch-clamp analysis and neural excitability studies. A brief schematic or concept diagram elucidating the experimental setup could aid comprehension.

In both Methodology sections: patch clamp and NES we added brief diagrams, and also modified patch clamp methodology description to make it more informative   

  • Results should be presented with a focus on key findings, facilitating easier interpretation for readers.

In the result section only key results have been finally highlighted

  • The language used is generally clear but can benefit from a more concise and structured approach. Ensure that each sentence contributes directly to the understanding of the study. Clarify certain technical terms and abbreviations for the broader readership.

The text was generally redrafted with an attempt to simplify the terminology and shorten certain paragraps – especially in the discussion section

  • Consider incorporating a conceptual diagram illustrating the experimental design. This can enhance the paper's accessibility and aid in understanding complex methodologies.

It is an important remark and we strongly agree that the general concept of the study should be more clearly presented.  We rewrote introduction describing concept of the paper in points. Finally, after discussions we didn’t decide do add the general scheme to avoid repeating text with the figure, but hope that with introduction modification the general concept will be clear to reader.

  • The paper alludes to investigating genetic causes for clinical variability. Expanding on this aspect, perhaps by providing additional context or potential hypotheses, could enrich the genetic analysis section.Some recent papers should be discussed in the discussion section: https://pubmed.ncbi.nlm.nih.gov/37704745/ ; https://pubmed.ncbi.nlm.nih.gov/37759906/ ; https://pubmed.ncbi.nlm.nih.gov/33923061/

We are grateful to the Reviewer for these suggestions – all these papers were discussed in both patch clamp and WES finding sections and added to the references.

Reviewer 2 Report

Comments and Suggestions for Authors

At the manuscript “Functional Characteristics of the Nav1.1 p.Arg1596Cys Mutation Associated with Varying Severity of Epilepsy Phenotypes” by Drs. Witkowski et al authors investigated mutations of the SCN1A gene, encoding the voltage-dependent Na+ channel’s alpha subunit which is associated with diverse epileptic activity and seizures.  Authors aimed of to describe properties of mutated channel and investigate genetic causes for clinical syndromes’ variability using exome sequencing SCN1A-associated phenotypes. To assess the impact of mutations it was used patch-clamp analysis and in vivo neural excitability studies.

 It was demonstrated that in cells expressing the mutant channel, sodium currents were reduced, that is significantly new and important result. Also it was shown increased excitability of peripheral motor neurons in mutation carriers that is quite understandable.

  Authors concluded that p.Arg1596Cys substitution inhibits channel function, affecting steady-state inactivation and inactivation kinetics and its clinical manifestations involve epileptic symptoms and increased excitability of peripheral motor neurons.

 I have no fundamental complaints: the research was carried out carefully, the methods were adequate and the research task seemed clear. However, I have a number of questions that the authors may not be able to answer.

 The authors write (LINE 285) that persistently weakened transmission may activate compensatory mechanisms that increase the excitability of motor neurons in order to maintain their activity at an appropriate level. This is logical, is there any indirect evidence to support this? In particular, increased neuronal activity is often associated with increased expression of Activity Regulated Cytoskeleton Associated Protein (Arc) (f.e. Sibarov et al Front Neurol. 2023; doi: 10.3389/fneur.2023.1201104) or William A Catterall, Franck Kalume, and John C Oakley; NaV1.1 channels and epilepsy J Physiol. 2010 Jun 1; 588(Pt 11): 1849–1859. doi: 10.1113/jphysiol.2010.187484; PMCID: PMC2901973; PMID: 20194124

Perhaps the authors have some thoughts on this issue and it would be interesting to know the authors' opinion.

 Since, as the authors rightly pointed out, clinical manifestations are largely determined by a subpopulation of mutated neurons, is it possible to use highly selective antiepileptic drugs targeting only a certain population of neurons? Is it possible to create specific antibodies for mutant Nav1 channels?

 The presentation of a subject is systematic and comprehensive and analysis is proper. I am happy to recommend the manuscript for the publication after minor corrections mentioned above.

Author Response

Dear Reviewer,

We would like to thank you for all valuable remarks. We applied many changes text based on the above suggestions, being absolutely sure that this will make our paper clearer and will add value to it.

Here are detailed responses to Reviewers’ remarks:

  • The authors write (LINE 285) that persistently weakened transmission may activate compensatory mechanisms that increase the excitability of motor neurons in order to maintain their activity at an appropriate level. This is logical, is there any indirect evidence to support this? In particular, increased neuronal activity is often associated with increased expression of Activity Regulated Cytoskeleton Associated Protein (Arc) (f.e. Sibarov et al Front Neurol. 2023; doi: 10.3389/fneur.2023.1201104).

Indeed, it is a valuable concept and we added paragraphs discussing this paper to the NES section in Discussion

  • Since, as the authors rightly pointed out, clinical manifestations are largely determined by a subpopulation of mutated neurons, is it possible to use highly selective antiepileptic drugs targeting only a certain population of neurons? Is it possible to create specific antibodies for mutant Nav1 channels?

We are grateful for this remark which encouraged us to add additional paragraphs to discussion path-clamp section, describing the novel approaches and treatment concepts focused on correction of the mutation consequences  (line 276).